# Add and Go: FRET Acceptor for Live-Cell Measurements Modulated by Externally Provided Ligand

**DOI:** 10.3390/ijms23084396

**Published:** 2022-04-15

**Authors:** Alexey S. Gavrikov, Nina G. Bozhanova, Mikhail S. Baranov, Alexander S. Mishin

**Affiliations:** 1Shemyakin-Ovchinnikov Institute of Bioorganic Chemistry, Russian Academy of Sciences, Miklukho-Maklay St. 16/10, 117997 Moscow, Russia; gavrikovalexey1@gmail.com (A.S.G.); nbozhanova@gmail.com (N.G.B.); baranovmikes@gmail.com (M.S.B.); 2Pirogov Russian National Research Medical University, Ostrovitianov 1, 117997 Moscow, Russia

**Keywords:** FRET, fluorogen-activating proteins, FAPs

## Abstract

A substantial number of genetically encoded fluorescent sensors rely on the changes in FRET efficiency between fluorescent cores, measured in ratiometric mode, with acceptor photobleaching or by changes in fluorescence lifetime. We report on a modulated FRET acceptor allowing for simplified one-channel FRET measurement based on a previously reported fluorogen-activating protein, DiB1. Upon the addition of the cell-permeable chromophore, the fluorescence of the donor-fluorescent protein mNeonGreen decreases, allowing for a simplified one-channel FRET measurement. The reported chemically modulated FRET acceptor is compatible with live-cell experiments and allows for prolonged time-lapse experiments with dynamic energy transfer evaluation.

## 1. Introduction

The design of many molecular probes used for studying protein–protein interactions in living cells is based on Förster resonance energy transfer (FRET) [1,2,3,4]. Still, an accurate measurement of FRET efficiency *in cellulo* remains a challenging task that requires a set of laborious control measurements, such as photobleaching of the acceptor with intense light [5]. Other approaches for FRET quantification, including fluorescence lifetime imaging, spectral imaging, and fluorescence polarization imaging, are far less accessible for researchers and/or require expensive hardware [6].

Modulated FRET acceptors provide an alternative way of measuring FRET. These acceptors are capable of switching between two states with different absorption properties, effectively turning FRET “on” and “off”. Until recently, the list of modulated FRET acceptors has been limited to just a few photochromic dyes and photochromic fluorescent proteins (FPs) [7,8].

Fluorogen-activating proteins (FAPs) are a group of nonfluorescent proteins that do not absorb in the visible wavelength area by themselves. Instead, they acquire these properties upon binding to an externally provided ligand [9,10,11]. Therefore, in theory these probes could be used as modulated FRET acceptors. The advantages of transient binders as FRET acceptors were discussed [10], but to the best of our knowledge have only been shown once using lifetime measurements [12]. Here, we report the use of a FAP (DiB1) as a modulated genetically encoded FRET acceptor for intensity-based FRET measurement. DiB1 is a truncated (~20.2 kDa) and mutated variant of bacterial lipocalin (Blc). In bacteria, Blc is involved in the transfer of fatty acids. The DiB1 protein was engineered to reversibly bind small fluorogens [11]. We evaluated the performance of the FRET pair consisting of conventional FP mNeonGreen [13] as a donor and a bipartite acceptor, activated by the transient interaction of the protein (DiB1) and small-molecule (compound **3** from Ref. [14]) parts. We estimate that the FRET efficiency of this donor–acceptor pair reaches ~45% in live-cell experiments.

## 2. Results and Discussion

First, we assessed the performance of the recently published protein:fluorogen complex [14] as a FRET acceptor. The protein part of the reporter, DiB1, transiently binds a cell-permeable chromophore (compound **3** from [14], (Z)-4-(4-(diethylamino)-2-(difluoroboryl)-benzylidene)-1H-imidazo [1,2-a]pyridin-5(4H)-one, Figure 1A), resulting in a complex with an absorbance maximum at 605 nm (Figure 1B). This compound is similar to BODIPY, which is actively used in FRET applications [15,16,17]. We also tested the FRET efficiency in pairs DiB2 and DiB3 with compound **3**. We fused DiB1 with mNeonGreen, a bright monomeric green fluorescent protein previously reported as an efficient FRET donor [18], and measured the fluorescence spectra of the purified protein solution with or without compound **3**. Comparing the fluorescence emission of the donor before and after the addition of compound **3**, we estimated FRET efficiency between mNeonGreen FP and the DiB1:**3** complex as ~61% (Figure 1C). This value agrees with the calculated Förster radius (~5.45 nm). As an additional control, we performed an enzymatic cleavage of mNeonGreen-DiB1 fusion protein with proteinase K after the addition of compound **3**, resulting in the restoration of fluorescence to the initial value (Figure 1C). FRET pairs with DiB2 and DiB3 showed very low FRET efficiency (~5–10%, data not shown) and were excluded from further experiments.

We further tested the one-channel FRET readout of the mNeonGreen:DiB1:**3** FRET pair in live HeLa cells. We chose histone H2B fusions as a model system due to the convenience of nuclear localization and the slow exchange rate of this histone [19]. Therefore, the ratio of fluorescence signal after and before the activation of the DiB1 acceptor could be reliably determined. During the experiment, we followed the fluorescence intensity of mNeonGreen in cells transiently transfected with the H2B-mNeonGreen-DiB1 construct (Figure 1D). The initial level of fluorescence signal served as a baseline, corresponding to nil FRET efficiency. Upon the addition of compound **3** to the imaging media, we observed a sharp decline in the fluorescence signal, plateauing at ~55% of the initial intensity (Figure 1E, Appendix A). In contrast, no fluorescence quenching was observed in control cells expressing H2B-mNeonGreen construct, which lacks the modulated FRET acceptors (Figure 1E, Appendix A). Therefore, the FRET efficiency of the mNeonGreen-DiB1 pair in living cells estimated by single-channel measurement was equal to ~45%. We also conducted an additional control experiment in which the donor and the acceptor were fused with non-interacting proteins (H2B and vimentin, respectively). In this case, the fluorescence of the donor was not influenced by the addition of compound **3**, as expected (Appendix A).

Importantly, the fluorescence of the donor immediately before the activation of the DiB1:**3** acceptor serves as an internal control. Therefore, the full dynamic range of the response could be extracted from single-channel imaging data without laborious control experiments, ratiometric imaging, or acceptor photobleaching.

In principle, the transient nature of DiBs:ligand interactions allows for repetitive FRET measurements by sequential addition and washing off the ligand over the course of an experiment. We tested this technique in live HeLa cells transiently transfected with H2B-mNeonGreen-DiB1 fusion protein (Figure 2A; Appendix A). With a simple perfusion system, we observed almost complete washing of compound **3** in our experiments in 4 min (with a flow rate of 120 μL/min through the slide). The activation of the acceptor after the addition of compound **3** at the same flow rate was faster; the fluorescence intensity of the donor dropped to a minimum in about one minute. The measured mean FRET efficiency in the repeated on–off cycles was about 45%. To ensure swift and complete staining without compromising temporal resolution, we recommend the use of compound **3** in excess (i.e., 3 × *K*_d_ or more). It should be noted that the fast dynamics (k_on_, k_off_) of the DiB1:**3** exchange remain unknown, hindering the design of precise dynamic FRET measurement at the molecular level.

We further assessed the applicability of the mNeonGreen-DiB1 FRET pair for studying protein–protein interactions. We labeled intermediate filaments (vimentin) simultaneously with vimentin-mNeonGreen and vimentin-DiB1. Due to the random incorporation of labeled monomers into the filaments, some vimentin monomers labeled with the donor will be found at a sufficient distance for energy transfer from the monomers labeled with the acceptor. In order to restrict the drift, these experiments were performed on fixed cells. We observed a mean FRET efficiency of 45% reaching the levels observed for donor–acceptor fusion in the previous experiment (Figure 2B,C). The observed FRET efficiency is in firm agreement with the vimentin filament structure of densely packed coiled coils: vimentin molecules form parallel dimers [20] and the C-terminus carrying DiB1 and mNeonGreen should be very close. We performed a similar experiment with cells transiently co-expressing H2B-mNeonGreen and H2B-DiB1 constructs (Figure 2D,E). The distance between nucleosomes can be small enough to allow for FRET [21]. In this case, FRET efficiency did not exceed 35% (mean = 21.5%).

Next, we evaluated the modulated FRET acceptor within the biosensors for the H3K9 trimethylation [22], and tension detection in focal contacts [2].

It is known that H3K9me3 modification is present mainly in heterochromatin. We measured the FRET efficiency with our H3K9me3 biosensor (Figure 3A) in living HeLa Kyoto cells in interphase when heterochromatin has this modification (Figure 3). We detected a relatively high efficiency of energy transfer in the interphase nuclei of cells, while the efficiency was unevenly distributed over the nuclei (Figure 3C,E). This distribution of FRET efficiency is in accordance with our expectations, since the H3K9me3 modification is located mainly in heterochromatin, which is heterogeneously distributed throughout the cell nucleus and can form various patterns.

Then, we used our modified tension sensor with a modulated acceptor to demonstrate the detection of the FRET efficiency over time (Figure 4). A living cell is constantly moving and with this tension sensor, it is easy to see changes in the FRET efficiency in focal contacts during cell movement. By alternately activating and deactivating the acceptor, we were able to see changes in the FRET efficiency in focal adhesions, and as a consequence, a change in tension during cell movement (Figure 4C, Appendix A, Appendix A).

Finally, we applied the modulated FRET acceptor to assess the protein–protein interaction of YAP1 and 14-3-3 [23] (Figure 5A). It is known that 14-3-3 locates diffusely in the cytoplasm, while the YAP1 protein is shuffling between the cytoplasm and the nucleus, depending on the phosphorylation, cell cycle, and other factors [24].

We labeled YAP1 and the 14-3-3 with mNeonGreen and DiB1, respectively. In live HeLa Kyoto cells we observed mNeonGreen-labeled YAP1 in the cytoplasm. Accordingly, we expected that after the addition of compound **3** and the activation of the acceptor, the FRET efficiency in the cytoplasm would be higher than in the nucleus. Indeed, the FRET efficiency in the cytoplasm turned out to be higher than in the nucleus (Figure 5B,D). In control experiments, in the absence of a modulated acceptor, the fluorescence intensity of the mNeonGreen did not change either in the cytoplasm or in the nucleus upon the addition of compound **3** (Figure 5B). We attribute the non-zero FRET efficiency in the nucleus to the background signal from the cytoplasm below and above the nucleus since the imaging was carried out under a widefield epifluorescence setup. To confirm this, we performed the same experiment under confocal microscopy conditions. We did not observe FRET in the nucleus, since the impact of the fluorescence from the cytoplasm was eliminated in a confocal regime (Figure 5E,F). To the best of our knowledge, the interaction of YAP1 with 14-3-3 is visualized in living HeLa Kyoto cells with FRET for the first time.

## 3. Materials and Methods

### 3.1. Molecular Cloning

Plasmids for mammalian expression encoding fusions of FP/FAP with histone H2B (H2B-DiB1, H2B-mNeonGreen, H2B-mNeonGreen-DiB1); vimentin (vimentin-DiB1, vimentin-mNeonGreen); YAP1 (YAP1-mNeonGreen); 14-3-3 (14-3-3-DiB1); plasmids encoding FP-FAP fusion with mNeonGreen-DiB1 for bacterial expression and in vitro measurements; H3K9me3 sensor; and tension sensor were assembled using Golden Gate cloning following MoClo standards [25,26,27]. Each transcriptional unit for mammalian expression consisted of the CMV promoter, coding sequence for the fusion protein, and the SV40 terminator. All Golden Gate cloning reactions were performed in the T4 ligase buffer (SibEnzyme, Novosibirsk, Russia) supplied with 10 U of T4 ligase, 20 U of either BsaI or BpiI (ThermoFisher, Waltham, USA), and 100 ng of each DNA fragment. Golden Gate reactions were performed with the following cycling conditions: 30 cycles between 37 °C and 16 °C (90 sec at 37 °C, 180 s at 16 °C).

### 3.2. Protein Expression and Purification

The fusion protein mNeonGreen-DiB1 was cloned into pBAD/His vector and expressed in Top10 *E. coli* cells. One bacterial colony was inoculated into 200 mL of LB broth and grown to OD600 = 0.1 at 37 °C. Expression of the recombinant proteins was induced with L-arabinose (to a final concentration of 7 mM). After 16 h at 37 °C, cells were harvested and resuspended in 5 mL of PBS buffer (pH 7.4, VWR chemicals). Cells were destroyed by sonication and centrifuged to obtain cell-free lysates. Proteins were purified using TALON metal affinity resin (Clontech). Resin was washed with 15–20 volumes of PBS. Then, resin beads were washed with PBS supplemented with 0.1M EDTA (pH 7.4) to elute proteins. Finally, the protein samples were dialyzed against PBS (pH 7.4).

### 3.3. Fluorescence Spectra and Förster Radius Evaluation

Fluorescence emission spectra were recorded on an Agilent Cary Eclipse fluorescence spectrophotometer. Spectra were acquired without and in the presence of compound 3 (final concentration—1 µM). The typical protein concentration was 100 nM. Enzymatic cleavage was performed with proteinase K (2.5 U). Förster radius was computed using python script from [28].

### 3.4. Cell Culture and Transient Transfection

HeLa cells were grown in Dulbecco’s modified Eagle’s medium (DMEM) (PanEco, Moscow, Russia) supplied with 50 U/mL penicillin and 50 µg/mL streptomycin (PanEco, Moscow, Russia), 2 mM L-glutamine (PanEco, Moscow, Russia), and 10% fetal bovine serum (HyClone, Thermo Scientific, Waltham, USA) at 37 °C and 5% CO_2_. For transient transfection, FuGENE HD reagent (Promega, Madison, USA) was used. A standard protocol was used to set up transfection; the DNA:reagent mixture was incubated for 20 min prior to the addition to the cells. Typically, 1 μg of DNA and 3 μL of the reagent were used. Transfection was performed in a serum-free OptiMEM (PanEco, Moscow, Russia). Immediately before imaging, DMEM was replaced with Hanks’ Buffer (PanEco, Moscow, Russia) supplemented with 20 mM HEPES (Sigma, Darmstadt, Germany).

### 3.5. Cell Culture Fixation

Before imaging, cell cultures cotransfected with H2B-mNeonGreen and H2B-DiB1, or vimentin-mNeonGreen and vimentin-DiB1s, were fixed with 2% (*w*/*v*) paraformaldehyde (Merck, Darmstadt, Germany) and washed four times with Hanks’ buffer supplemented with 20 mM HEPES.

### 3.6. Widefield Fluorescence Microscopy and Washing

Widefield fluorescence microscopy was performed with Leica 6000, Leica HCX PL APO 100X/1.40–0.70NA oil immersion objective, Zyla sCMOS camera (Andor, Oxford, UK), and CoolLED pE-300 light source. A GFP cube filter set (Leica, Wetzlar, Germany) was used (excitation filter: 470/40, dichroic mirror: 500, suppression filter: 525/50). Typical illumination power ranged from 1 to 6 W/cm^2^ with exposure times ranging from 50 to 150 ms. Experiments with repeated FRET measurements were performed with live cells cultured on µ–Slide I 0.4 Luer ibiTreat (Ibidi, Gräfelfing, Germany) slides. The alternating flow of HHBS (Hanks’ buffer with 20 mM Hepes) or HHBS supplemented with compound **3** (3 µM) through a µ–Slide was driven by a custom-made syringe pump injection system controlled by Raspberry Pi 3. Flow speed was set to 120 µL/min, providing a full exchange of the media in the µ–Slide in ~1 min.

### 3.7. Confocal Microscopy

Confocal imaging was performed using TSC SP2 (Leica Microsystem, Germany) based on an inverted fluorescent microscope Leica DM IRE equipped with HCX PL APO Lbd.BL 63 × 1.40 oil lens and an argon laser. Images were acquired with the excitation by the 488 nm line of the argon laser (≈9 µW), with the emission collected at 500–550 nm.

### 3.8. FRET Efficiency Measurements

The widefield epifluorescence images before and after the addition of compound **3** were aligned in FIJI [29] software to correct for cell movement, followed by the background subtraction and 2x downsampling. Finally, the image acquired after acceptor activation was divided (pixelwise) by the image before acceptor activation. The FRET efficiency value was calculated by the subtraction of the resulting ratio image from 1.

## 4. Conclusions

Ligand-exchanging fluorogen-activating proteins, such as DiB1, provide robust temporal control over protein labeling. In this work, we showed the utility of DiB1 as a modulated FRET acceptor with in-place reference. In contrast to the irreversible bipartite labeling systems, such as HaloTag and SNAP-Tag [30], labeling with exchangeable tags allows for fast on-demand staining, improved photostability, and multiplex high-content imaging [31]. The changes in the fluorescence signal in sequential staining–washing cycles provided the estimate of FRET in time-lapse experiments. In the future, our system can be improved by finding a “dark” chromophore as an acceptor activator, freeing up an additional imaging channel. Moreover, the combined use of DiB1 and other orthogonal exchangeable protein–fluorogen labeling systems, such as FAST [10,12], will add additional dimensions for the multiplex functional imaging of cellular processes. 

## Figures and Tables

**Figure 1 ijms-23-04396-f001:**
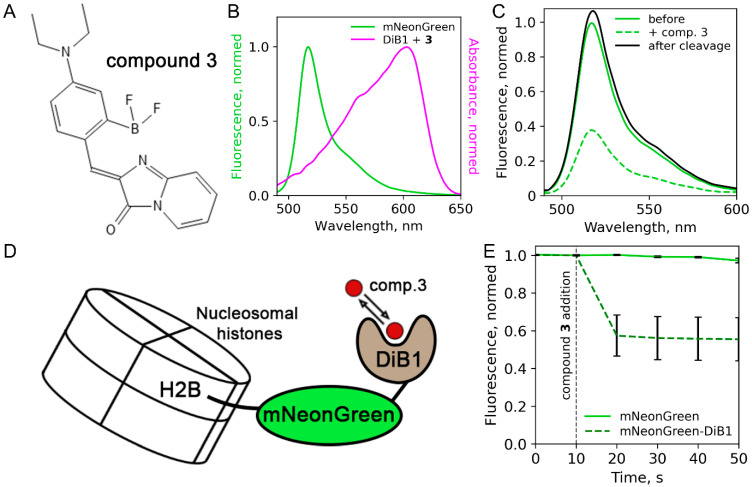
FRET efficiency measurements in vitro and in living cells. (**A**) Structure of compound **3**. (**B**) Spectral overlap between the fluorescence emission spectrum of mNeonGreen and the absorbance spectrum of the DiB1:**3** complex. (**C**) mNeonGreen emission spectra before the compound **3** addition (green line), after the compound 3 addition to the final concentration of 1 μM (dashed green line), and after the enzymatic cleavage (black line). (**D**) Scheme of the H2B-mNeonGreen-DiB1 used in live-cell experiment. (**E**) Fluorescence intensity of mNeonGreen in live HeLa cells transiently transfected with H2B-mNeonGreen (green solid line) or H2B-mNeonGreen-DiB1 (green dashed line); vertical dashed line indicates the addition of compound **3** to the final concentration of 3 μM; error bars represent standard deviation (technical replicates = 23).

**Figure 2 ijms-23-04396-f002:**
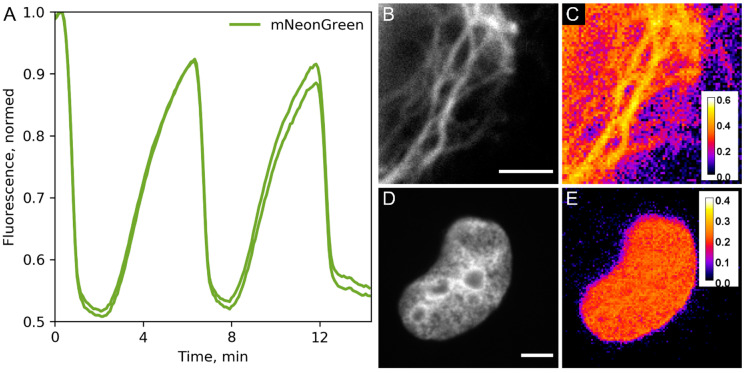
Possible applications of the reported FRET pair. (**A**) Repeating FRET activation in live HeLa cells. Green lines represent average intensities of mNeonGreen (donor) fluorescence in labeled nuclei of live HeLa cells transiently transfected with H2B-mNeonGreen-DiB1 construct (each line corresponds to a different cell nucleus). (**B**–**E**) Studying protein–protein interactions. (**B**,**C**) Transient co-transfection with vimentin-mNeonGreen and vimentin-DiB1 constructs, fixed HeLa cells. (**B**) Widefield image of vimentin fibers labeled with mNeonGreen. (**C**) FRET image of the same vimentin fibers; pseudocolor indicates FRET efficiency after the addition of 3 μM compound **3**. (**D**,**E**) Transient co-transfection with H2B-mNeonGreen and H2B-DiB1 constructs, fixed HeLa cells. (**D**) Widefield image of H2B-mNeonGreen. (**E**) FRET image of the same nucleus; pseudocolor indicates FRET efficiency after the addition of 3 μM compound **3**. Scale bars are 3 μm (vimentin) and 5 μm (H2B).

**Figure 3 ijms-23-04396-f003:**
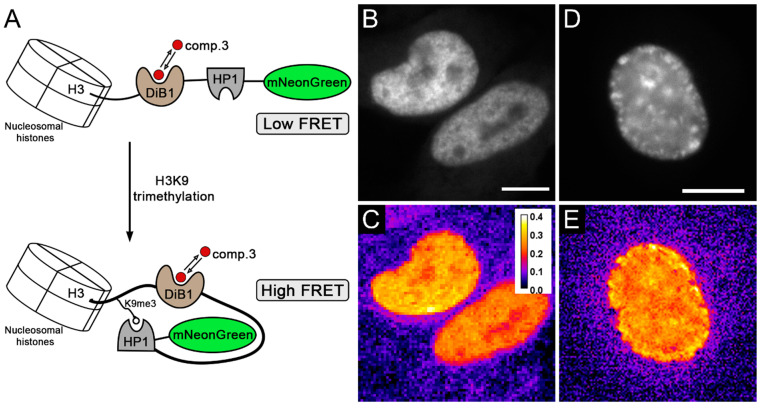
Modified biosensor for H3K9 trimethylation evaluation based on a new FRET pair with the modulated acceptor. (**A**) Scheme of the new H3K9me3 biosensor and its working mechanism. (**B**,**D**) Widefield image of live HeLa Kyoto cells transiently transfected with modified H3K9me3 biosensor construct; scale bar is 5 μm. (**C**,**E**) Visualization of energy transfer in live HeLa Kyoto cells; pseudocolor indicates FRET efficiency after the addition of compound **3** (3 μM). Panels C and E share the same color scale of FRET efficiency.

**Figure 4 ijms-23-04396-f004:**
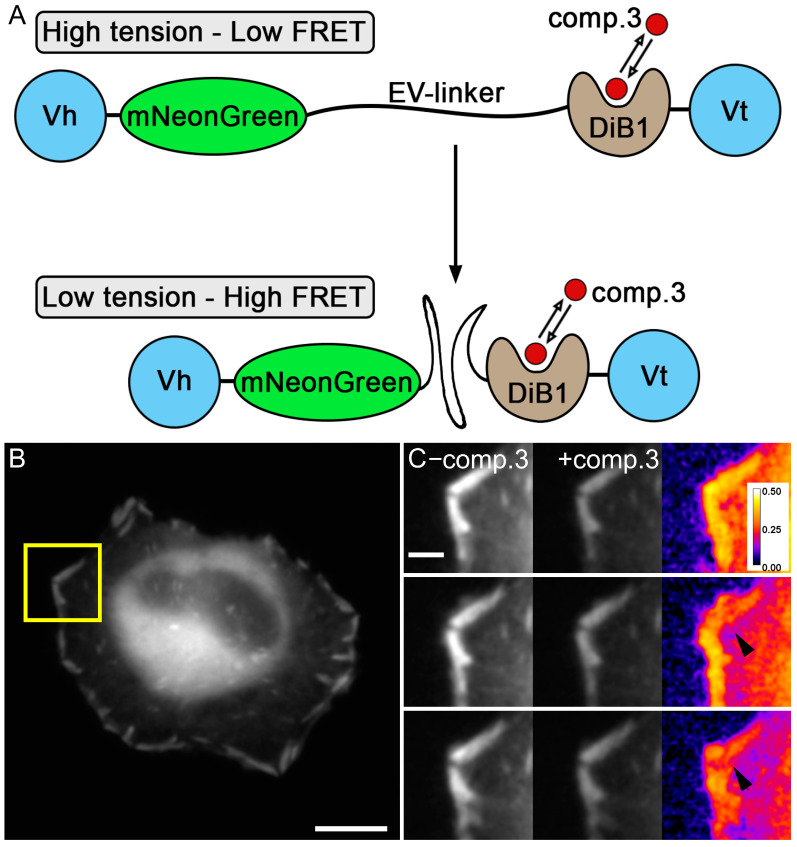
Modified tension sensor based on a new FRET pair with the modulated acceptor. (**A**) Scheme of the new tension sensor and its working mechanism. (**B**) Widefield image of live HeLa Kyoto cells transiently transfected with modified tension sensor construct; scale bar is 20 μm. (**C**) Widefield images of the region indicated by a yellow rectangle on panel (**B**) before, after compound **3** addition, and FRET efficiency images of the same region over time (pseudocolor—FRET efficiency); scale bar is 3 μm.

**Figure 5 ijms-23-04396-f005:**
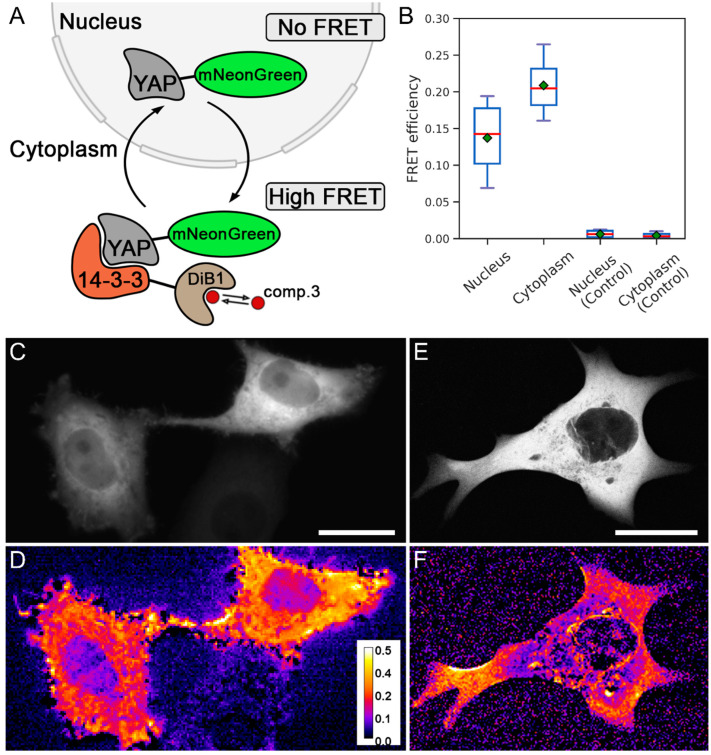
Detection of the interaction of YAP1 and 14-3-3 proteins. (**A**) Scheme of labeled proteins’ localization and their interaction. (**B**) FRET efficiency in different cell compartments (nucleus and cytoplasm) in experimental live HeLa Kyoto cells transiently cotransfected with two constructs, YAP1-mNeonGreen and 14-3-3-DiB1, and in live HeLa Kyoto cells transiently transfected with construct YAP-mNeonGreen. Red lines represent median values, green rectangles represent mean values (technical replicates = 11). (**C**) Widefield image of live HeLa Kyoto cells transiently cotransfected with YAP1-mNeonGreen and 14-3-3-DiB1 constructs. (**D**) Visualization of energy transfer in live HeLa Kyoto cells from panel (**C**); pseudocolor indicates FRET efficiency after the addition of 3 μM compound **3**. (**E**) Confocal image of live HeLa Kyoto cells transiently cotransfected with YAP1-mNeonGreen and 14-3-3-DiB1 constructs. (**F**) Visualization of energy transfer in live HeLa Kyoto cells from panel (**E**); pseudocolor indicates FRET efficiency after the addition of 3 μM compound **3** (panels D and F share the same color scale of FRET efficiency); scale bars are 20 μm.

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
