# Peer review of "Add and Go: FRET Acceptor for Live-Cell Measurements Modulated by Externally Provided Ligand"

_ijms, 2022, doi:10.3390/ijms23084396_

Round 1

Reviewer 1 Report

Dear Editor

-This paper can be accepted after some additions. 

-Are the authors sure of the structure of compound 3 in Figure 1?

-The level of English re-view for all manuscript. 

-The authors gave only 24 references. They should be increased on FRET and fluorescent compounds just as A Bodipy-bearing pillar [5] arene for mimicking photosynthesis: Multi-fluorophoric light harvesting system, Tetrahedron Letters 59 (20), 1958-1962. Synthesis and photophysical properties of modifiable single, dual, and triple-boron dipyrromethene (Bodipy) complexes Tetrahedron Letters 56 (14), 1873-1877. On/off rhodamine-BODIPY-based fluorimetric/colorimetric sensor for detection of mercury (II) in half-aqueous medium, IEEE Sensors Journal 19 (6), 2009-2015.

Best Regards

Author Response

-This paper can be accepted after some additions. 

-Are the authors sure of the structure of compound 3 in Figure 1?

RESPONSE: Yes, we confirm that the structure is correct. We have redrawn the panel for clarity. We follow one of the acceptable conventions for BF2-containing compounds, e.g. Ref. 14

-The level of English re-view for all manuscript. 

RESPONSE: Thank you. We have carefully proofread the text and have made the corrections.

-The authors gave only 24 references. They should be increased on FRET and fluorescent compounds just as A Bodipy-bearing pillar [5] arene for mimicking photosynthesis: Multi-fluorophoric light harvesting system, Tetrahedron Letters 59 (20), 1958-1962. Synthesis and photophysical properties of modifiable single, dual, and triple-boron dipyrromethene (Bodipy) complexes Tetrahedron Letters 56 (14), 1873-1877. On/off rhodamine-BODIPY-based fluorimetric/colorimetric sensor for detection of mercury (II) in half-aqueous medium, IEEE Sensors Journal 19 (6), 2009-2015.

RESPONSE: Thank you for the suggestion. We have added suggested references (Refs. 19-21)

Reviewer 2 Report

Review report

Add and Go: FRET acceptor for live-cell measurements modulated by externally provided ligand

By: Alexey S. Gavrikov, Nina G. Bozhanova, Mikhail S. Baranov, Alexander S. Mishin

In this manuscript, Gavrikov et al. present an interesting application of a Fluorogen-activating Protein (FAP), DiB1, as a FRET acceptor for cellular imaging. DiB1 reversibly binds an exogenous ligand that can be introduced to and removed from the sample through perfusion. Upon binding to DiB1, the ligand optical properties change and its absorption of light increases. This protein has been presented and characterized in previous work from the same group. In the current manuscript, they fuse DiB1 to a fluorescent protein, mNeonGreen, that is commonly used as a donor in Förster resonance energy transfer (FRET) constructs. The configuration presented here has the main advantage of not requiring the preparation of multiple samples for the calculation of FRET efficiency: imaging in the absence of a FRET acceptor can be performed before introducing the fluorogenic molecule binding to DiB1, and this provides a readout of the fluorescence of the donor by itself. Upon addition of the fluorogenic molecule, FRET with the acceptor, the DiB1:ligand complex, is established. Removal of the fluorogenic molecule allows for verification of the fluorescence of the donor post-FRET imaging and for repeated measurements on the same sample at different points in time. The authors show the working principle of the system in purified protein at first, then move on to in-cell imaging of several constructs to showcase the compatibility of the FRET system with this type of investigations. They choose five constructs to measure FRET in static as well as dynamic conditions. The concept and ideas presented here have a good degree of originality, however the construct is not completely new: as the authors themselves point out in the introduction, a similar application was shown in ref. 12 (Benaissa et al, Nature Communications, 2021) albeit only using lifetime-based measurements, and as part of a larger showcase of how FAPs can be applied in live cell imaging. There is, however a certain degree of overlap between the experiments and applications presented by Benaissa et al. and the submitted manuscript. Nevertheless, Gavrikov et al. are using a new protein, with different features from the FAST protein from Benaissa et al. The authors should leverage this fact and present the advantages of DiB1 compared to FAST, and the orthogonal/complementary applications that this new protein could be used for: all this should be explicitly mentioned in the introduction of the paper, referencing both their previous work (DiB1was characterized in earlier papers) and the content of the submitted manuscript. Furthermore, the authors embarked in a thorough characterization of the usage of DiB1 as FRET acceptor with a suite of molecular constructs, which is definitely useful information for potential users who would like to employ DiB1 in their assays.

Overall, the manuscript submitted is of interest for the field, but the quality, presentation and scientific rigor must be improved for publication. As it is now, the paper is not suitable for publication and several issues need to be addressed. Here below, please find a more detailed list of comments.

Major issues that must be addressed before considering for publication:

  1. More information should be provided on DiB1 in the introduction, to give context to readers not familiar with the protein. More detailed disclosure of previous characterization should be presented, to understand the choice of DiB1:compound 3 for this work.
  2. The authors decided to use compound 3 from their previous work as fluorogenic ligand: in Bozhanova et al, IJMS, 2018 (ref 14) they characterized the spectral and binding properties of this molecule to DiB1, and one key properties that is not mentioned at all in the submitted manuscript is that DiB1:compound3 is fluorescent. Why did the author not measure any of the sensitized emission from DiB1:3 in this manuscript? This measurement could corroborate the FRET observed through donor quenching. The author should also comment on why they chose DiB1 instead of DiB2 or DiB3. From the data in ref 14, it appears that DiB2:3 and DiB3:3 are non-fluorescent complexes, making them dark acceptors. A dark acceptor would open up a usable channel for multiplexing fluorescence imaging in the spectral region above 600nm. This would greatly increase the impact of the molecular system proposed here.
    1. The sensitized emission of DiB1:3 should be measures in purified protein as well as in cell imaging conditions, to provide a reference of its magnitude
  3. There is no indication on how the FRET efficiency was calculated throughout the paper: a mention to that and a detailed report on how images were analyzed to produce the figures would make the manuscript clearer and help other researchers understand the methodology for reproduction purposes. Throughout the text, the authors should be consistent in reporting one parameter only for the comparisons of data (i.e. FRET efficiency), whereas now they alternate between efficiencies and % decrease in intensity which makes it difficult to follow.
    1. With regard to the data in Figure 1, panel E, they state a decrease of fluorescence of 37% and declare a FRET efficiency of 63%. The calculations do not seem to match: the authors should check if this is a typo, or they should explain how they reach this conclusion
    2. Overall, if FRET is the metric measured and intended for use, it should be reported for all the data and figures. Instead of reporting the relative fluorescence (e.g.: Figure 2 panel C&E, Figure 3 panel C, Figure 4 panel C right most insets, Figure 5 panel C) FRET values, as efficiency, should be presented.
  4. The Kd of compound 3 for DiB1 is ~90nM and in ref 14, for imaging they use submicromolar concentrations, however in the presented work the concentration used for FRET experiments ranges between 1 and 3 m No explanation is provided for this. A titration of compound 3 and the effect of its concentration is highly recommended for the data in vitro and for at least one of the in cellulo systems to assess the useful range of concentrations for these experiments.
  5. The experiments in vitro are lacking a control where the emission of mNeonGreen alone is shown in the presence of compound 3 when DiB1 is not in the construct. A titration is highly recommended.
  6. For the experiments with the H2B fusion constructs, no images are provided for reference. An image sequence or a movie representative of the data shown in Figure 1E should be provided in the supplementary material.
    1. Figure 2A would actually make more sense if grouped with Figure 1, since it is related to the H2B experiment, and not the vimentin constructs
    2. The additional experiments where H2B-mNeonGreen and H2B-DiB1 constructs are used to show intermolecular FRET should also be grouped together with the H2B data.
    3. The authors should present a control experiment where mNeonGreen and DiB1 are fused to molecules not interacting with one another.
  7. The experiments with the H2B fusion constructs and with the H3K9me3 sensor should be presented sequentially one after the other, as they both image proteins that localize in the nucleus. They also present increasingly complex system: H2B data show a proof of concept of imaging, whereas the H3K9me3 data show the applicability of DiB1 in FRET-based sensors.
  8. The H3K9me3 data spark interest because DiB1 is used in a sensor application, however the data lack information that would increase the impact of the manuscript: in Figure 3 cells at interphase are shown with fluorescence and FRET efficiency images reported, but there is no comparison with cells not at interphase. Is the DiB1 sensor capable of visualizing the difference? The statement “…while the efficiency was unevenly distributed over the nuclei (Figure 174 3C)” should be explained, because it is not clear whether it is an expected outcome or not, and what is its relevance for the DiB1-based sensor presented here.
  9. Application of DiB1 in the tension sensor is interesting, the authors show some data however they are limited:
    1. In the main text of the paper, Figure 4 provides a good idea of what is being measured, but more examples of FRET changes over time should be provided. A larger dataset than panel C is advised.
    2. A movie showing both the raw fluorescence changes and the FRET changes should be provided as supplementary material.
  10. The experiments where mNeonGreen and DiB1 are used to look at YAP and 14-3-3 interactions are interesting, yet a more complete dataset and thorough examination is required, especially because this is the first time where the YAP/14-3-3 interaction is visualized through FRET, according to the authors. Raw fluorescence images should be provided: on line 201 is reported “… we observed mNeonGreen-labeled YAP in the cytoplasm.”, there is no mention of the levels in the nucleus and no data are provided.
    1. The authors should try to obtain confocal images of cells to calculate the actual FRET efficiency in the nucleus without contribution from the cytoplasm.
    2. Application of the FRET system to verify a physiologically relevant hypothesis would strengthen the paper and provide evidence for its usability beyond proof of principle. An interesting feature of the YAP/14-3-3 complex and its response to hypoxia is presented by Jia et al., Oncogenesis, 2019 (doi:10.1038/s41389-019-0143-1), where simple co-localization was used but a FRET-based approach would have provided a more direct measure.
  11. A very important point that the authors should address is the kinetic limitation of this system: DiB1 is a FRET acceptor that relies on the dynamic binding and unbinding of compound 3 to work, which is advantageous for some aspects (e.: photostability) but it sets a constraint on the time resolution. When looking at dynamic changes in time of FRET, what is the expected best possible time resolution of a construct using DiB1:compound 3 as an acceptor? Kinetics of the ligand exchange should be taken into account, and if possible, an experimental demonstration should be provided.
  12. Materials and Methods are not detailed enough that reproducibility by others could be assured. Sequences, or source, or GenBank accession number of the parts of the constructs should be made available. Furthermore:
    1. Details of the protein expression and purification are insufficient and should be expanded.
    2. Transfection parameters should be reported as well. The authors should claim whether they have investigated or not the effect of transfection conditions on the outcome of the experiments: does the use of low/high amounts of DNA or transfection reagents affect the observed outcome?
    3. The parameters of imaging need more details: what are the excitation/emission center wavelengths and bandwidth of the GFP cube? Not all GFP cubes from different companies are the same. What are the parameters used for imaging? Exposure time, power density, and so on? What is the image analysis workflow used (background subtraction, any pre-processing, quantification)? This is important because a quantitative metric is to be extracted from the data (FRET efficiency) and it can be used to quantify biological processes.
  13. The authors should clearly specify the statistics of their data by indicating the number of times they replicated their experiments, and whether these are technical or biological replicates. The authors did not provide any document containing supplementary material, aside of one video: they are invited to submit more representative images and data for each of the experimental evaluations they present, as indicated above in my previous points, and as they see fit to support the claims they make in the manuscript.

Minor issues to be addressed before considering for publication:

  • The authors should revise the manuscript more thoroughly for example: they left, in the current version, parts of the template document (see lines 117-119). An overall grammatical check before resubmission is advised.
  • Figure preparation quality should be improved:
    • g.: in most figures, grey lines around the plots are visible but surround only some sides of the plot. The authors should format the figures in a more organized and cleaner way: if borders are to be used to divide the space, they should be consistent and well organized.
    • Scale bars and color bars should clearly indicate the scale and what measurements they refer to.
    • Figure 2, panel A: it should be clearly stated what the two different lines represent. Are they different cells, different ROIs in the same cells, different samples? Please specify
    • When average data from multiple experiments are shown, the number of replicates should be clearly and consistently reported. It should be specified if these are technical, or biological replicates.
  • The conclusions of the manuscript are currently weak and do not provide a good overview of the potential of the system used. The authors should elaborate on the advantages over currently used proteins of equivalent (g.: FAST) or similar (e.g.: SNAPtag, HaloTag) nature, on limitations and how future studies and development could improve a FRET system based on DiB1.

Author Response

Major issues 

  1. More information should be provided on DiB1 in the introduction, to give context to readers not familiar with the protein. More detailed disclosure of previous characterization should be presented, to understand the choice of DiB1:compound 3 for this work.

RESPONSE: Thank you for your suggestion on improving clarity of the manuscript. We added a brief description of the DiB1  protein in the Introduction. We also explained the rationale behind the selection of  DiB1:compound 3 in the Results section.

  1. The authors decided to use compound 3 from their previous work as fluorogenic ligand: in Bozhanova et al, IJMS, 2018 (ref 14) they characterized the spectral and binding properties of this molecule to DiB1, and one key properties that is not mentioned at all in the submitted manuscript is that DiB1:compound3 is fluorescent. Why did the author not measure any of the sensitized emission from DiB1:3 in this manuscript? This measurement could corroborate the FRET observed through donor quenching. The author should also comment on why they chose DiB1 instead of DiB2 or DiB3. From the data in ref 14, it appears that DiB2:3 and DiB3:3 are non-fluorescent complexes, making them dark acceptors. A dark acceptor would open up a usable channel for multiplexing fluorescence imaging in the spectral region above 600nm. This would greatly increase the impact of the molecular system proposed here.
  1. The sensitized emission of DiB1:3 should be measures in purified protein as well as in cell imaging conditions, to provide a reference of its magnitude

RESPONSE: Thank you for the comment. We agree that a perfect dark acceptor can free up a channel (red) for imaging. Unfortunately, acceptors with DiB2 and DiB3 exhibited very low FRET efficiency of about just 5-10% during the in vitro screening. Due to such a small dynamic range of fluorescence change, we excluded DiB2 and DiB3 from further experiments. The reasons for this less-than-expected FRET performance of DiB2:compound 3 and DiB3: compound 3 complexes remain to be elucidated and fall out of the scope of current Manuscript. 

On the matter of sensitized emission, we refrained from classical sensitized emission measurements due to the lack of a suitable filter set capable of eliminating channel bleed-through on our microscope. While we fully agree that sensitized emission in vitro can additionally corroborate the FRET measured via donor quenching, in this revision we include an additional experimental evidence: measuring FRET before and immediately after enzymatic cleavage of the donor-acceptor fusion protein (see updated figure 1C).

  1. There is no indication on how the FRET efficiency was calculated throughout the paper: a mention to that and a detailed report on how images were analyzed to produce the figures would make the manuscript clearer and help other researchers understand the methodology for reproduction purposes. Throughout the text, the authors should be consistent in reporting one parameter only for the comparisons of data (i.e. FRET efficiency), whereas now they alternate between efficiencies and % decrease in intensity which makes it difficult to follow.

RESPONSE:  Thank you. We now include a detailed explanation of FRET calculation within the Methods section (2.8). As suggested, we now use ‘FRET efficiency’ in all figures.

  1. With regard to the data in Figure 1, panel E, they state a decrease of fluorescence of 37% and declare a FRET efficiency of 63%. The calculations do not seem to match: the authors should check if this is a typo, or they should explain how they reach this conclusion

RESPONSE: Thank you!  We corrected the text in accordance with the data shown on Figure 1E. The correct FRET efficiency is 45%.

b. Overall, if FRET is the metric measured and intended for use, it should be reported for all the data and figures. Instead of reporting the relative fluorescence (e.g.: Figure 2 panel C&E, Figure 3 panel C, Figure 4 panel C right most insets, Figure 5 panel C) FRET values, as efficiency, should be presented.

RESPONSE: Fixed. We updated all the figures in question (2, 3, 4 and 5). 

  1. The Kd of compound 3 for DiB1 is ~90nM and in ref 14, for imaging they use submicromolar concentrations, however in the presented work the concentration used for FRET experiments ranges between 1 and 3 m No explanation is provided for this. A titration of compound 3 and the effect of its concentration is highly recommended for the data in vitro and for at least one of the in cellulo systems to assess the useful range of concentrations for these experiments.

RESPONSE: Thank you for this comment. It seems that we really exceed the minimum necessary concentration of compound 3. To address this concern we performed titration in cellulo, as suggested (Figure S1). However, from our experience with transient labeling, we would recommend the use of slight excess of the compound 3 (i.e. 3*Kd) to ensure swift and complete staining. 

  1. The experiments in vitro are lacking a control where the emission of mNeonGreen alone is shown in the presence of compound 3 when DiB1 is not in the construct. A titration is highly recommended.

RESPONSE: Thank you. We believe, that the in cellulo control (1E, green solid line) together with updated 1C (black solid line – after cleavage) provide sufficient experimental support for the lack of influence of compound 3 on the fluorescence of mNeonGreen.

  1. For the experiments with the H2B fusion constructs, no images are provided for reference. An image sequence or a movie representative of the data shown in Figure 1E should be provided in the supplementary material.

RESPONSE: Thank you. We included a new Supplementary figure S2 with the sequence of reference images, as suggested. 

  1. Figure 2A would actually make more sense if grouped with Figure 1, since it is related to the H2B experiment, and not the vimentin constructs
  2. The additional experiments where H2B-mNeonGreen and H2B-DiB1 constructs are used to show intermolecular FRET should also be grouped together with the H2B data.

RESPONSE: Thank you for the suggestion to reorder figures. We tried it (suggestions a and b), and based on the response of our peers, decided to keep the original layout. We follow the logic that firstly there is an experiment with fusion protein, then experiments with intermolecular FRET within one intracellular localization, and finally the sensor and protein-protein interaction. Now, the figures are presented in the order of gradual increase in the complexity of experiments.

c. The authors should present a control experiment where mNeonGreen and DiB1 are fused to molecules not interacting with one another.

RESPONSE: Thank you. As suggested, we now include a control experiment l (Figure S3) with mNeonGreen and DiB1 fused to non-interacting molecules (H2B and vimentin, respectively).

  1. The experiments with the H2B fusion constructs and with the H3K9me3 sensor should be presented sequentially one after the other, as they both image proteins that localize in the nucleus. They also present increasingly complex system: H2B data show a proof of concept of imaging, whereas the H3K9me3 data show the applicability of DiB1 in FRET-based sensors.

RESPONSE: Thanks for the suggestion to change the order of the figures. As we wrote in the response to points 6a,b, we follow the same logic of increased complexity, but in a slightly different manner. 

  1. The H3K9me3 data spark interest because DiB1 is used in a sensor application, however the data lack information that would increase the impact of the manuscript: in Figure 3 cells at interphase are shown with fluorescence and FRET efficiency images reported, but there is no comparison with cells not at interphase. Is the DiB1 sensor capable of visualizing the difference? The statement “…while the efficiency was unevenly distributed over the nuclei (Figure 174 3C)” should be explained, because it is not clear whether it is an expected outcome or not, and what is its relevance for the DiB1-based sensor presented here.

RESPONSE: Thank you for your inspiring comment. Indeed, we intend to follow the entire cell cycle with sensors of similar design in a systematic manner in the future, but we feel that thorough study of different patterns of H3K9me3 signal with relation to the cell cycle falls out of the scope of the current manuscript. For the purpose of this revision, we now include an additional image of a nucleus with distinct patterns of the H3K9me3 signal (Figure 3 D,E). 

  1. Application of DiB1 in the tension sensor is interesting, the authors show some data however they are limited:
  1. In the main text of the paper, Figure 4 provides a good idea of what is being measured, but more examples of FRET changes over time should be provided. A larger dataset than panel C is advised.
  2. A movie showing both the raw fluorescence changes and the FRET changes should be provided as supplementary material.

RESPONSE: Thank you for your request for additional data. As suggested, we expanded the set of images (Supplementary figure S4) and included a movie (Supplementary movie 2)  showing both the raw fluorescence and the FRET signal.

  1. The experiments where mNeonGreen and DiB1 are used to look at YAP and 14-3-3 interactions are interesting, yet a more complete dataset and thorough examination is required, especially because this is the first time where the YAP/14-3-3 interaction is visualized through FRET, according to the authors. Raw fluorescence images should be provided: on line 201 is reported “… we observed mNeonGreen-labeled YAP in the cytoplasm.”, there is no mention of the levels in the nucleus and no data are provided.
  1. The authors should try to obtain confocal images of cells to calculate the actual FRET efficiency in the nucleus without contribution from the cytoplasm.
  2. Application of the FRET system to verify a physiologically relevant hypothesis would strengthen the paper and provide evidence for its usability beyond proof of principle. An interesting feature of the YAP/14-3-3 complex and its response to hypoxia is presented by Jia et al., Oncogenesis, 2019 (doi:10.1038/s41389-019-0143-1), where simple co-localization was used but a FRET-based approach would have provided a more direct measure.

RESPONSE: Thank you for these comments. As suggested, we now include fluorescence data on Figure 5 (panels C,E). As suggested, we added confocal images to Figure 5 too (panels E,F). We fully agree that experiments on the physiology and translocation of YAP pose a considerable interest.  However, we feel that detailed study of this interesting biological topic falls out of the scope of the current manuscript, which is more focused on the method itself.

  1. A very important point that the authors should address is the kinetic limitation of this system: DiB1 is a FRET acceptor that relies on the dynamic binding and unbinding of compound 3 to work, which is advantageous for some aspects (e.: photostability) but it sets a constraint on the time resolution. When looking at dynamic changes in time of FRET, what is the expected best possible time resolution of a construct using DiB1:compound 3 as an acceptor? Kinetics of the ligand exchange should be taken into account, and if possible, an experimental demonstration should be provided.

RESPONSE: Thank you for your valuable comments on the kinetic limitation of the system. The temporal resolution can be roughly estimated from the experimental data on Figure 2A. As can be seen from the distance between the peaks, the temporal resolution or all practical means and purposes, is about 1 FRET frame per ~6 minutes. This is how long it takes to stain the sample with  the compound 3 and wash it off in our perfusion system. Since the washing off compound 3 appears to be considerably slower than staining, it limits the overall time resolution. While we expect that miniaturized perfusion systems with lower dead volumes can achieve faster buffer exchange, we do not expect that the temporal resolution of the system can be drastically improved. 

  1. Materials and Methods are not detailed enough that reproducibility by others could be assured. Sequences, or source, or GenBank accession number of the parts of the constructs should be made available. Furthermore:
  1. Details of the protein expression and purification are insufficient and should be expanded.
  2. Transfection parameters should be reported as well. The authors should claim whether they have investigated or not the effect of transfection conditions on the outcome of the experiments: does the use of low/high amounts of DNA or transfection reagents affect the observed outcome?
  3. The parameters of imaging need more details: what are the excitation/emission center wavelengths and bandwidth of the GFP cube? Not all GFP cubes from different companies are the same. What are the parameters used for imaging? Exposure time, power density, and so on? What is the image analysis workflow used (background subtraction, any pre-processing, quantification)? This is important because a quantitative metric is to be extracted from the data (FRET efficiency) and it can be used to quantify biological processes.

RESPONSE: Thanks for your comments. We have expanded the Methods section as requested and added the missing information. We now also provide protein sequences as supplementary information

  1. The authors should clearly specify the statistics of their data by indicating the number of times they replicated their experiments, and whether these are technical or biological replicates. The authors did not provide any document containing supplementary material, aside of one video: they are invited to submit more representative images and data for each of the experimental evaluations they present, as indicated above in my previous points, and as they see fit to support the claims they make in the manuscript.

RESPONSE: Thank you, we have added information about replicates. We also expanded the supplementary information  and included relevant data.

Minor issues to be addressed before considering for publication:

  • The authors should revise the manuscript more thoroughly for example: they left, in the current version, parts of the template document (see lines 117-119). An overall grammatical check before resubmission is advised.

RESPONSE: Fixed.

  • Figure preparation quality should be improved:
    • g.: in most figures, grey lines around the plots are visible but surround only some sides of the plot. The authors should format the figures in a more organized and cleaner way: if borders are to be used to divide the space, they should be consistent and well organized.
    • Scale bars and color bars should clearly indicate the scale and what measurements they refer to.
    • Figure 2, panel A: it should be clearly stated what the two different lines represent. Are they different cells, different ROIs in the same cells, different samples? Please specify
    • When average data from multiple experiments are shown, the number of replicates should be clearly and consistently reported. It should be specified if these are technical, or biological replicates.

RESPONSE: All done. The information about scale bars and color bars are included in the legends.

  • The conclusions of the manuscript are currently weak and do not provide a good overview of the potential of the system used. The authors should elaborate on the advantages over currently used proteins of equivalent (g.: FAST) or similar (e.g.: SNAPtag, HaloTag) nature, on limitations and how future studies and development could improve a FRET system based on DiB1.

RESPONSE: We extended the conclusion, as suggested.

Round 2

Reviewer 2 Report

In this new version of the manuscript, the authors addressed most of the concerns and comments that arose during the first revision. They added very valuable experiments and provided a more complete set of data with expanded supplementary information and movies showing a more complete picture of the capabilities of their proposed FRET system. The Figures and presentation were greatly improved, and the presentation is now coherent and cohesive. Although during my first revision I proposed some further experimental investigations to broaden the impact and increase the scope of the work, the authors gracefully explained how these might go beyond the aim of their current manuscript and I acknowledge and accept their reasoning as compelling.

A few minor points are still standing and I would kindly ask the authors to address them. Please see here below my final comments. Once these are clarified, I believe that the manuscript is suitable for publications.

Thanks to the authors for sharing the results of their research, and for their positive reception of comments and request for edits throughout the peer-review process.

Authors:

On the matter of sensitized emission, we refrained from classical sensitized emission measurements due to the lack of a suitable filter set capable of eliminating channel bleed-through on our microscope. While we fully agree that sensitized emission in vitro can additionally corroborate the FRET measured via donor quenching, in this revision we include an additional experimental evidence: measuring FRET before and immediately after enzymatic cleavage of the donor-acceptor fusion protein (see updated figure 1C).

Reply:

I appreciate the response from the authors, and I understand the rationale behind that. The additional experiment provided (enzymatic cleavage) definitely corroborates the FRET between the donor and acceptor in this construct. The technical inability to measure sensitized emission may pose a challenge to the practical realization, but I think that the authors make it clear that there is sensitized emission when discussing the results, not only in the conclusions, so that potential users of this construct (DiB1:Compound 3) are aware of this experimental fact.

Authors:

Thank you for this comment. It seems that we really exceed the minimum necessary concentration of compound 3. To address this concern we performed titration in cellulo, as suggested (Figure S1). However, from our experience with transient labeling, we would recommend the use of slight excess of the compound (i.e. 3*Kd) to ensure swift and complete staining.

Reply:

The titration provided in Figure S1 is extremely useful, thanks for including it in the manuscript. Please add a mention to Figure S1 in the main text, and include the suggestion to “use of slight excess of the compound (i.e. 3*Kd) to ensure swift and complete staining” in the manuscript.

Authors:

Thank you for your valuable comments on the kinetic limitation of the system. The temporal resolution can be roughly estimated from the experimental data on Figure 2A. As can be seen from the distance between the peaks, the temporal resolution or all practical means and purposes, is about 1 FRET frame per ~6 minutes. This is how long it takes to stain the sample with the compound 3 and wash it off in our perfusion system. Since the washing off compound appears to be considerably slower than staining, it limits the overall time resolution. While we expect that miniaturized perfusion systems with lower dead volumes can achieve faster buffer exchange, we do not expect that the temporal resolution of the system can be drastically improved.

Reply:

Thank you for pointing out the kinetics limitation of the system, and for adding a sentence about it in the manuscript. What the authors are referring to here, are mainly the staining/de-staining kinetics which are really important for sample manipulation and for switching on and off the FRET system. Likely these are going to be of main interest for most users and they are addressed in the manuscript.

The concern that I was trying to raise in my first revision is mainly connected to the microscopic exchange of ligand (Compound 3) at the molecular level: one can imagine that DiB1 constantly “binds-and-release” Compound 3 with a kon (concentration dependent) and koff , from which a ton and toff could be calculated. My guess is that these are in the order of few to few hundred milliseconds: these on and off events at the molecular level may limit the use of a DiB1:Compound3 acceptor for investigating fast molecular dynamics that sometimes are investigated using FRET systems (e.g.: rapid molecular motions, ion-sensing and so on). I was interested and curious to know if the authors have thought about it and could add a brief comment in their manuscript to acknowledge this limitation.

Some remaining minor issues:

  • Figures: the issue with grey lines around the plots seem to be still there, please clean up the figures so that they have a more homogenous appearance. It might be an issue of the version that I received, but hopefully the appearance will be fixed in the published version.
  • Figure 3 (C,E), Figure 5 (D,F): please mention that the panels share the same color scale of FRET intensity, for clarity purpose.
  • Materials and Methods, Protein expression and purification: please mention the type of plasmid and protein induction conditions. The protocol for eluting protein from a TALON resin column using EDTA does not seem common, I wonder if it is a typo or if there is a reason for using EDTA instead of imidazole. The authors should specify why they use it, as EDTA would also strip their resin of the metal ion and this would end up in the final protein solution, potentially interfering with their FRET experiments.

Author Response

The titration provided in Figure S1 is extremely useful, thanks for including it in the manuscript. Please add a mention to Figure S1 in the main text, and include the suggestion to “use of slight excess of the compound 3 (i.e. 3*Kd) to ensure swift and complete staining” in the manuscript.
RESPONSE:
Thank you. In this revision, the supplementary Figure S1 is referenced immediately after the reference to Figure 1E. We have included the recommended concentration of compound 3 two paragraphs below, in the section discussing the repetitive FRET measurements with a perfusion system.

Thank you for pointing out the kinetics limitation of the system, and for adding a sentence about it in the manuscript. What the authors are referring to here, are mainly the staining/de-staining kinetics which are really important for sample manipulation and for switching on and off the FRET system. Likely these are going to be of main interest for most users and they are addressed in the manuscript.

The concern that I was trying to raise in my first revision is mainly connected to the microscopic exchange of ligand (Compound 3) at the molecular level: one can imagine that DiB1 constantly “binds-and-release” Compound 3 with a kon (concentration dependent) and koff , from which a ton and toff could be calculated. My guess is that these are in the order of few to few hundred milliseconds: these on and off events at the molecular level may limit the use of a DiB1:Compound3 acceptor for investigating fast molecular dynamics that sometimes are investigated using FRET systems (e.g.: rapid molecular motions, ion-sensing and so on). I was interested and curious to know if the authors have thought about it and could add a brief comment in their manuscript to acknowledge this limitation.
RESPONSE:
Thank you. We agree with the raised concern. Unfortunately, the kon and koff have not been directly measured for any DiB:fluorogen pair. We have only partial evidence of the ton from super-resolution imaging (Ref 14) of DiB1:compound 3, i.e. 16ms exposure time was empirically found to be sufficient to engulf the majority of bright binding events. However, it would be inappropriate to draw any conclusions from this data, since the observed single-molecule bursts are subject to the ‘survival bias’: we register only the best events, and many more conformation-brightness combinations may be present within the system.  We have acknowledged the lack of knowledge on the exchange kinetics at single-molecule level and potential limitations of the system at the end of the section discussing the repetitive FRET measurements.  

Some remaining minor issues:
Figures: the issue with grey lines around the plots seem to be still there, please clean up the figures so that they have a more homogenous appearance. It might be an issue of the version that I received, but hopefully the appearance will be fixed in the published version.
RESPONSE:
Thank you. We suspect this to be a compression glitch in the manuscript tracking system. We will again check for grey lines in proofs. 

Figure 3 (C,E), Figure 5 (D,F): please mention that the panels share the same color scale of FRET intensity, for clarity purpose.
RESPONSE:
Thank you for your comment. As suggested, we included the mention that panels Figure 3 (C,E) and Figure 5 (D,F) share the same color scale of FRET intensity.

Materials and Methods, Protein expression and purification: please mention the type of plasmid and protein induction conditions. The protocol for eluting protein from a TALON resin column using EDTA does not seem common, I wonder if it is a typo or if there is a reason for using EDTA instead of imidazole. The authors should specify why they use it, as EDTA would also strip their resin of the metal ion and this would end up in the final protein solution, potentially interfering with their FRET experiments.
RESPONSE.
Thank you for your comment. We amended the methods section. On the topic of EDTA elution: we routinely use EDTA elution for TALON resin in accordance with the manufacturer’s instructions, to ensure complete removal of the protein from the resin. In this way,  we use less resin: due to abundance and high molar extinction coefficients of colored and fluorescent proteins in our lab, we refrain from reuse of the same resin. Indeed, the missing text was the obligatory final dialysis (PBS 7.4) step, similar to the original DiB purification protocol (Ref. 11).